



**A simulation of the large-scale drifting snow storm in a turbulent**
**boundary layer**
**Zhengshi Wang[1,2], Shuming Jia [1,2*]**
[1] State Key Laboratory of Aerodynamics, China Aerodynamic Research and
Development Center, Mianyang Sichuan 621000, China
[2] Computational Aerodynamics Institute, China Aerodynamics Research and
Development Center, Mianyang, Sichuan 621000, China
[*] **Corresponding:** Shuming Jia (jiashm17@cardc.cn)



**Abstract.** Drifting snow storm is an important aeolian process that reshapes alpine
glaciers and polar ice shelves, and it may also affect the climate system and
hydrological cycle since flying snow particles exchange considerable mass and energy
with air flow. Prior studies have rarely considered the full-scale drifting snow storm in
the turbulent boundary layer, thus, the transportation feature of snow flow higher in
the air and its contribution are largely unknown. In this study, a large eddy simulation
is combined with a subgrid scale velocity model to simulate the atmospheric turbulent
boundary layer, and a Lagrangian particle tracking method is adopted to track the
trajectories of snow particles. A drifting snow storm that is hundreds of meters in
depth and exhibits obvious spatial structures is produced. The snow transport flux
profile at high altitude, previously not observed, is quite different from that near the
surface, thus, the extrapolated transport flux profile may largely underestimate the
total transport flux. At the same time, the development of a drifting snow storm
involves three typical stages, the rapid growth, the gentle growth and the equilibrium
stages, in which the large-scale updrafts and subgrid scale fluctuating velocities
basically dominate the first and second stage, respectively. This research provides an
effective way to get an insight into natural drifting snow storms.



## 1 Introduction

Snow, one type of solid precipitation, is an important sources of material to mountain glaciers and polar ice sheets, which are widespread throughout high and cold regions (Chang et al., 2016; Gordon and Taylor, 2009; Lehning et al., 2008). A common natural phenomenon over snow cover is the drifting snow storm, which occurs when the wind speed exceeds a critical value (Doorschot et al., 2004; Li and Pomeroy, 1997; Sturm and Stuefer, 2013). Drifting snow can entrain loose snow particles on the bed into the air, which may be further transported to high altitude by turbulent eddies (King, 1990; Mann et al., 2000; Nemoto and Nishimura, 2004). Drifting snow clouds typically can range in thickness from tens to thousands of meters (Mahesh et al., 2003; Palm et al., 2011), which may not only affect people's daily life by reducing the visibility and producing local accumulation (Gordon and Taylor, 2009; Mohamed et al., 1998) , but also can influence the global climate system evolution by changing the mass and energy balance of ice shelves (Cess and Yagai, 1991; Hanesiak and Wang, 2005; Hinzman et al., 2005; Lenaerts and Broeke, 2012).

Several field experiments on drifting snow storm have been performed (Bintanja, 2001; Budd, 1966; Dingle and Radok, 1961; Doorschot et al., 2004; Gallée et al., 2013; Gordon and Taylor, 2009; Guyomarch et al., 2014; Kobayashi, 1978; Mann et al., 2000; Nishimura and Nemoto, 2005; Nishimura et al., 2015; Pomeroy and Gray, 1990; Sbuhei, 1985; Schmidt, 1982; Sturm and Stuefer, 2013) since the middle of the last century. However, the measurements are commonly conducted near the surface, thus, the drifting snow features at high altitude are unknown, and the impacts of these



features are difficult to assess. A thorough investigation documenting the evolution
process and structure of a full-scale drifting snow storm is essential to understand this
natural phenomenon and assess its impacts.
Drifting snow models, on the other hand, offer a panoramic view of the evolution
process of drifting snow and thus have become one of the most useful research
approaches. Many continuum medium models of drifting snow (Bintanja, 2000; Déry
and Yau, 1999; Schneiderbauer and Prokop, 2011; Uematsu et al., 1991; Vionnet et al.,
2013) have advanced the knowledge of natural drifting snow to a great extent.
However, a particle-tracking drifting snow model is still needed since the particle
characteristics and its motion require further investigation. Although a series of
particle tracking models (Huang et al., 2016; Huang and Shi, 2017; Huang and Wang,
2015; 2016; Nemoto and Nishimura, 2004; Zhang and Huang, 2008; Zwaaftink et al.,
2014) have been established, these models have generally focused on the grain-bed
interactions and particle motions near the surface. Thus, a drifting snow model aimed
at producing a large-scale drifting snow storm in a turbulent boundary layer deserves
further exploration.
In this study, a drifting snow model in the atmospheric boundary layer that focuses
on the full-scale drifting snow storm is established. The wind field is solved using a
large eddy simulation for the purpose of generating a turbulent atmospheric boundary
layer. A subgrid scale (SGS) velocity is also considered to include the diffusive effect
of small scale turbulence. Finally, particle motion is calculated using a Lagrangian
particle tracking method. The large-scale drifting snow storm is produced and its

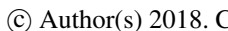



spatial structures and transport features are analyzed.

## 2    Model and methods

### 2.1    Simulation of a turbulent atmospheric boundary layer

The mesoscale atmosphere prediction pattern ARPS (Advanced Regional Prediction
System, version 5.3.3) is adopted to simulate the turbulent atmospheric boundary
layer, in which the filtered three-dimensional compressible non-hydrostatic
Naiver-Stokes equation is solved (Xue et al., 2001):
$$\frac{\partial \rho}{\partial t} + \frac{\partial}{\partial x_i}(\rho \tilde{u}_i) = 0 \tag{1}$$

$$\frac{\partial \rho \tilde{u}_i}{\partial t} + \frac{\partial \rho u_i \tilde{u}_j}{\partial x_j} = -\frac{\partial \tilde{p}^*}{\partial x_i} + B\delta_{i3} - \frac{\partial \tau_{ij}}{\partial x_j} \tag{2}$$

where '~' represents variables that are filtered and the filtering scale is
$\tilde{\Delta} = \left(\Delta x_1 \Delta x_2 \Delta x_3\right)^{1/3}$, in which $\Delta x_i$ is the grid spacing along streamwise ($i = 1$),
spanwise ($i = 2$) and vertical direction ($i = 3$), respectively. $u_i$ is the instantaneous
wind speed component, and $x_i$ is the position coordinate. $t$ is time, $\delta_{ij}$ is the
Kronecker delta, $B = -g\,\rho'/\rho$ is the buoyancy caused by the air density perturbation
$\rho'$, and $g$ is the acceleration due to gravity. $p^* = p' - \alpha\nabla(\rho\mathbf{u})$ contains the pressure
perturbation term and damping term, where $\alpha$ is the damping coefficient and $\nabla$ is
the divergence. The subgrid stress $\tau_{ij}$ can be expressed as (Smagorinsky, 1963):
$$\tau_{ij} = -2\nu_t \tilde{S}_{ij} = -2\left(C_s \tilde{\Delta}\right)^2 \left|\tilde{S}\right| \tilde{S}_{ij} \tag{3}$$

where $\tilde{S}_{ij} = 0.5\left(\partial \tilde{u}_i/\partial x_j + \partial \tilde{u}_j/\partial x_i\right)$ is the strain rate tensor and $\left|\tilde{S}\right| = \sqrt{2\tilde{S}_{ij}\tilde{S}_{ij}}$, $C_s$
is Smagorinsky coefficient that is determined locally by the dynamic Lagrangian
model (Meneveau et al., 1996).



Considering the large grid spacing in simulating an atmospheric boundary layer
(where the information about turbulent vortices smaller than the grid size is missing),
the SGS velocity is also included. Namely, the local wind velocity $\tilde{u}_i\left(\vec{x}(t)\right)$ is
composed of a resolved Eulerian large-scale part $\tilde{u}_i\left(\vec{x}(t)\right)$ (obtained from the linear
weighting of surrounding grid points) and a fluctuating SGS contribution $u_i'(t)$. The
SGS velocity can be calculated from the SGS stochastic model of Vinkovic et al.

97      (2006):

$$du_i' = \left(-\frac{1}{T_L} + \frac{1}{2\tilde{k}}\frac{d\tilde{k}}{dt}\right)u_i'dt + \sqrt{\frac{4\tilde{k}}{3T_L}}d\eta_i(t) \qquad (4)$$

where $T_L = 4\tilde{k}\big/\left(3C_0\tilde{\varepsilon}\right)$ is the Lagrangian correlation time scale. Here, $C_0$ is the
Lagrangian constant, $\tilde{\varepsilon} = C_\varepsilon \tilde{k}^{3/2}\big/\tilde{\Delta}$ is the subgrid turbulence dissipation rate, $C_\varepsilon$ is
a constant, and $d\eta_i$ is the increment of a vector-valued Wiener process with zero
mean and variance $dt$. $\tilde{k}$ is the subgrid turbulent kinetic energy and can be obtained
from the transport equation (Deardorff, 1980):

$$\frac{\partial \tilde{k}}{\partial t} + \tilde{u}_j \frac{\partial \tilde{k}}{\partial x_j} = \frac{\nu_t}{3}\frac{g}{\theta_0}\frac{\partial \tilde{\theta}}{\partial x_3} + 2\nu_t \tilde{S}_{ij}^2 + 2\frac{\partial}{\partial x_j}\left(\nu_t \frac{\partial \tilde{k}}{\partial x_j}\right) + \tilde{\varepsilon} \qquad (5)$$

where $\theta$ is the potential temperature and $\theta_0$ is the surface potential temperature.
**2.2 Governing equation of particle motion**
The trajectory of each snow particle is calculated using a Lagrangian particle tracking
method. Since a snow particle has is almost $10^3$ times more dense than air, airborne
particles are assumed to process only gravity and fluid drag forces, and the governing
equations of particle motion can be expressed as (Dupont et al., 2013; Huang and
Wang, 2016; Vinkovic et al., 2006):



$$\frac{dx_{pi}}{dt} = u_{pi} \tag{6}$$

$$\frac{du_{pi}}{dt} = m_p \frac{V_{ri}}{T_p} f(Re_p) + \delta_{i3} g \tag{7}$$

where $x_{pi}$ and $u_{pi}$ are the position coordinate and velocity of the snow particle,
respectively. $m_p$ is the mass of the solid particle, $V_r$ is the relative speed between
the snow particle and air, and $T_p = \rho_p d_p^2 / 18 \rho v$ is the particle relaxation time, where
$\rho_p$ is the particle density, $d_p$ is the particle diameter and $v = 1.5e - 5$ is the
dynamic viscosity of air. $f(Re_p)$ can be expressed as (Clift et al., 1978):
$$f(Re_p) = \begin{cases} 1 & (Re_p < 1) \\ 1 + 0.15 Re_p^{0.687} & (Re_p \geq 1) \end{cases} \tag{8}$$

where $Re_p = V_r d / v$ is the particle Reynolds number.
**2.3 Initial conditions of snow particles**
To generate a large-scale drifting snow storm, a steady-state snow saltation condition
is set as the bottom boundary condition for particles. During drifting snow events, the
sum of residual fluid shear stress $\tau_f$ and particle-borne shear stress $\tau_p$ should be
equal to the total fluid shear stress $\tau$, thus, the particle-borne stress can be expressed
as:
$$\tau_p = \tau - \tau_f \tag{9}$$

Here, the residual fluid shear stress $\tau_f$ is set to be the threshold shear stress $\tau_{tf}$
of drifting snow, which can be read as (Clifton et al., 2006):
$$\tau_{tf} = A^2 g d (\rho_p - \rho) \tag{10}$$

in which $A = 0.2$ is a constant, $g$ is the gravity acceleration and $d$ is the mean
diameter of the snow particles.





At the same time, the particle-borne shear stress at the surface can be calculated
from the particle trajectories as (Nemoto and Nishimura, 2004):
$$\tau_p = \sum_{i=1}^{n_\downarrow} m_i u_{pi\downarrow} - \sum_{i=1}^{n_\uparrow} m_i u_{pi\uparrow} \qquad (11)$$

where $m_i$ is the mass of particle and $u_{pi\downarrow}$ and $u_{pi\uparrow}$ are the horizontal speeds of
impact and lift-off particles, respectively. $n_\downarrow$ and $n_\uparrow$ are the particle number per
unit area in unit time of impact and lift-off grains, respectively, which should be
equivalent in steady-state saltation. Thus, the number of lift-off particles per unit area
is:
$$n_\uparrow = n_\downarrow = \frac{\tau_p}{\langle m_i \rangle \left(1 - \langle e_h \rangle\right) \langle u_{pi\downarrow} \rangle} \qquad (12)$$

in which $\langle \; \rangle$ indicates the overall average, and $e_h$ is the horizontal restitution
coefficient of snow particle. According to Sugiura and Maeno (2000), the mean
horizontal restitution coefficient can be expressed as:
$$\langle e_h \rangle = \begin{cases} 0.48\theta_i^{0.01} & v_i \leq 1.27 ms^{-1} \\ 0.48\left(\dfrac{v_i}{1.27}\right)^{-\log\left(\frac{v_i}{1.27}\right)} \theta_i^{0.01} & v_i > 1.27 ms^{-1} \end{cases} \qquad (13)$$

where $\theta_i$ and $v_i$ are the impact velocity and angle, respectively. Here, $\theta_i$ has a
mean value of approximately 10° (Sugiura and Maeno, 2000), and $\langle v_i \rangle$ is set to be
the threshold of impact velocity, which is determined by setting ejection number
$n_e = 0.51 v_i^{0.6} \theta_i^{0.16}$ equal to 1. In this way, the mean horizontal velocity of impact
particles can be obtained through $\langle u_{pi\downarrow} \rangle = \langle v_i \rangle \cos \langle \theta_i \rangle$.
Then, the velocities of lift-off particles can be obtained from the restitution
coefficient of snow. The horizontal restitution coefficient obeys the normal

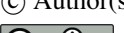



distribution with a mean value given in Eq. 13, and a standard variance as follow
(Sugiura and Maeno, 2000):
$$\sigma^2 = \begin{cases} 0.07\theta_i^{-0.06} & v_i \leq 0.52\,ms^{-1} \\ 0.07(\dfrac{v_i}{0.52})^{-\log(\frac{v_i}{0.52})}\theta_i^{-0.06} & v_i > 0.52\,ms^{-1} \end{cases} \qquad (14)$$

On the other hand, the vertical restitution coefficient can be described by a two
parameter gamma function (see Eq. 17), in which the parameter $\alpha$ and $\beta$ can be
expressed as (Sugiura and Maeno, 2000):
$$\alpha = \begin{cases} 1.22\theta_i^{0.47} & v_i \geq 0.84\,ms^{-1} \\ 1.22(\dfrac{v_i}{0.84})^{\log(\frac{v_i}{0.84})}\theta_i^{0.47} & 0.84 < v_i \leq 1.23\,ms^{-1} \\ 1.22(\dfrac{v_i}{0.84})^{\log(\frac{v_i}{0.84})}(\dfrac{v_i}{1.23})^{-2\log(\frac{v_i}{1.23})}\theta_i^{0.47} & v_i \geq 1.23\,ms^{-1} \end{cases} \qquad (15)$$

$$\beta = \begin{cases} 12.85\theta_i^{-1.41} & v_i \geq 0.84\,ms^{-1} \\ 12.85(\dfrac{v_i}{0.84})^{-\log(\frac{v_i}{0.84})}\theta_i^{-1.41} & 0.84 < v_i \leq 1.23\,ms^{-1} \\ 12.85(\dfrac{v_i}{0.84})^{-\log(\frac{v_i}{0.84})}(\dfrac{v_i}{1.23})^{\log(\frac{v_i}{1.23})}\theta_i^{-1.41} & v_i \geq 1.23\,ms^{-1} \end{cases} \qquad (16)$$

**2.4 Simulation details**
The computational domain is $1000 \times 500 \times 1000$ m, with a uniform horizontal grid
size of 5 m adopted to solve finer vortex structure in the atmospheric boundary layer.
The mean grid size in the vertical direction is 20 m, with a grid refinement algorithm
adopted near the surface (the finest grid size is 1 m). Periodic boundaries are used
along streamwise and spanwise dimensions, and the bottom is set as a grid wall. The
top is set as an open radiation boundary with a Rayleigh damping layer that is 250 m
in depth.





The atmosphere is neutral with an initial potential temperature of 300K, and an
initial relative humidity of 90%. The initial wind profile is logarithmic with a surface
roughness of 0.1m (Doorschot et al., 2004). Atmospheric turbulence is induced by
random initial potential temperature perturbations at the first-level grid level with a
maximum magnitude of 0.5K, and is sustained by a constant heat flux at the bottom.
The constant heat flux is 50 $Wm^{-2}$ according to the observation of Pomeroy and
Essery (1999).
For particles, periodic boundary conditions are also used at lateral boundaries, and
a rebound boundary condition without energy loss is adopted at the model top. The
bottom boundary condition for particles is given in Sect. 2.3, and is updated every 0.5
s. Additionally, each particle represents one particle parcel for the purpose of reducing
computational complexity.

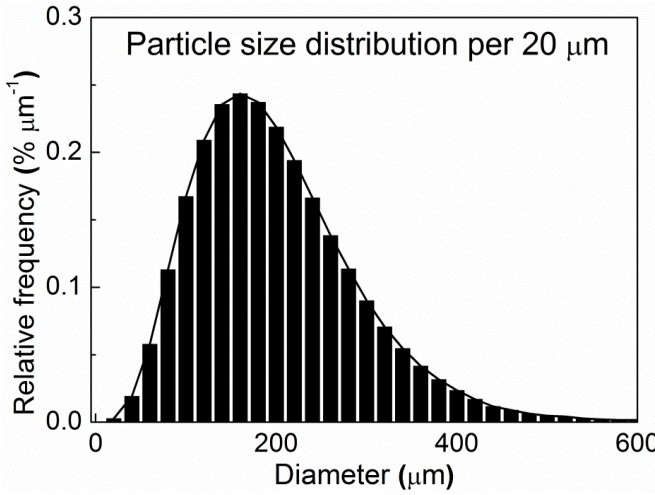


**Figure** 1. Size distribution of lift-off snow particles in this simulation.
The size distribution of lift-off particles in drifting snow can be well described by
the two-parameter gamma function (Budd, 1966; Gordon and Taylor, 2009;



Nishimura and Nemoto, 2005; Schmidt, 1982):
$$f(d) = \frac{d^{\alpha-1}}{\beta^{\alpha}\Gamma(\alpha)}\exp\left(-\frac{\beta}{d}\right) \tag{17}$$

where $d$ is the particle diameter, and $\alpha$ and $\beta$ are the shape and scale parameter of
the distribution, respectively. In this simulation, the diameters of lift-off snow
particles are given randomly from a gamma function with the parameters of $\alpha = 4$
and $\beta = 50$, as shown in Fig. 1, which is also consistent with observed particle size
distributions (Nishimura and Nemoto, 2005; Schmidt, 1982).

## 192    3    Results and discussions

### 193    3.1  Model validation

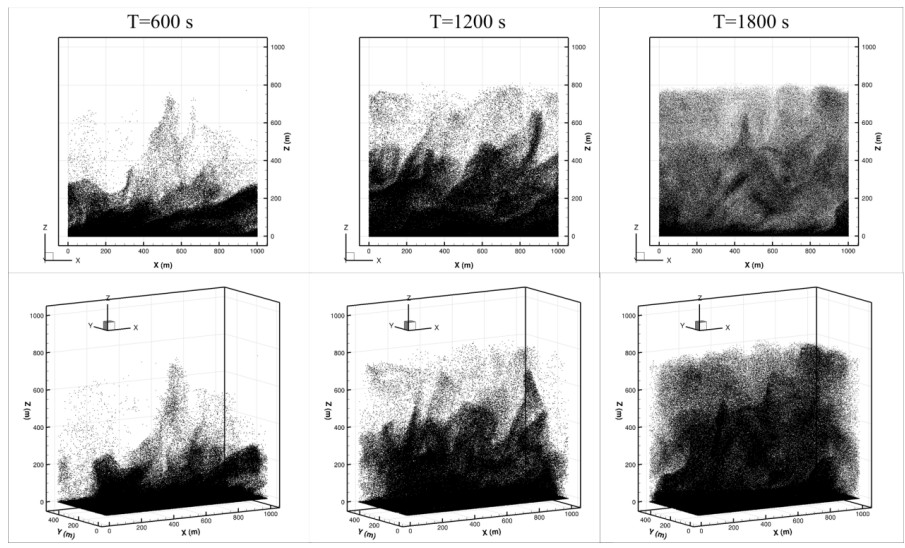


**Figure** 2. Drifting snow storm at different moments under the friction velocity of 0.29
ms[-1].
When drifting snow occurs in the atmospheric boundary layer, updrafts and
turbulence fluctuations can send snow particles to high altitude, forming a fully
developed drifting snow storm. Fig. 2 shows the drifting snow storm in the





atmospheric boundary layer at different moments, in which the friction velocity is
$u_* = 0.29 \ ms^{-1}$ and dark spots represent snow particles. It can be seen that drifting
snow storm experiences an evolution process from near the surface to high altitudes,
which induces the fact that particle concentration decreases along increasing height.
The high concentrations of drifting snow cloud are generally below 500 m, though
snow particles may reach up to approximately 800 m under this condition. This is also
consistent with observations (Mahesh et al., 2003; Palm et al., 2011).

Since a drifting snow storm exhibits a different structure from bottom to top, the

evolution of particle number density profile in the drifting snow storm is shown in Fig.
3, which is also compared with measurements of Mann et al. (2000) . From this figure,
the thickness of the drifting snow layer obviously increases with time, and almost
approaches its steady state after 1200 s. At the same time, the particle number density
basically decreases with height, which is consistent with the measurements of Mann
et al. (2000) at various friction velocities. The predicted particle number density at the
surface is much larger than at higher altitude and observations, mainly because the
saltating particles are also included.

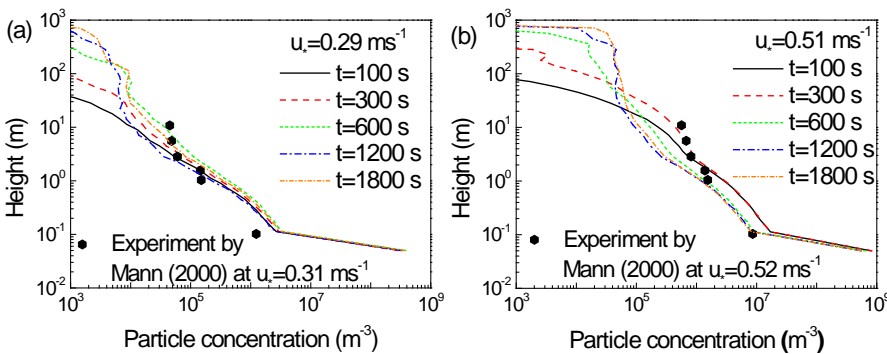


**Figure** 3. Evolution of particle number density under various friction velocities (a)



0.29 ms$^{-1}$ and (b) 0.51 ms$^{-1}$.
Generally, smaller particles are more likely to be transported higher in the air. Fig.
4 shows the variation of modeled average particle diameter versus height, which is
also compared with various field measurements (Nishimura and Nemoto, 2005;
Schmidt, 1982). Similar to the field observations, the average particle size basically
decreases with height at lower altitude but is almost constant above 1 m. The average
particle diameter is approximately 75 μm ranging from one meter to hundreds of
meters in height, which is also consistent with the measurements of K Nishimura and
Nemoto (2005).

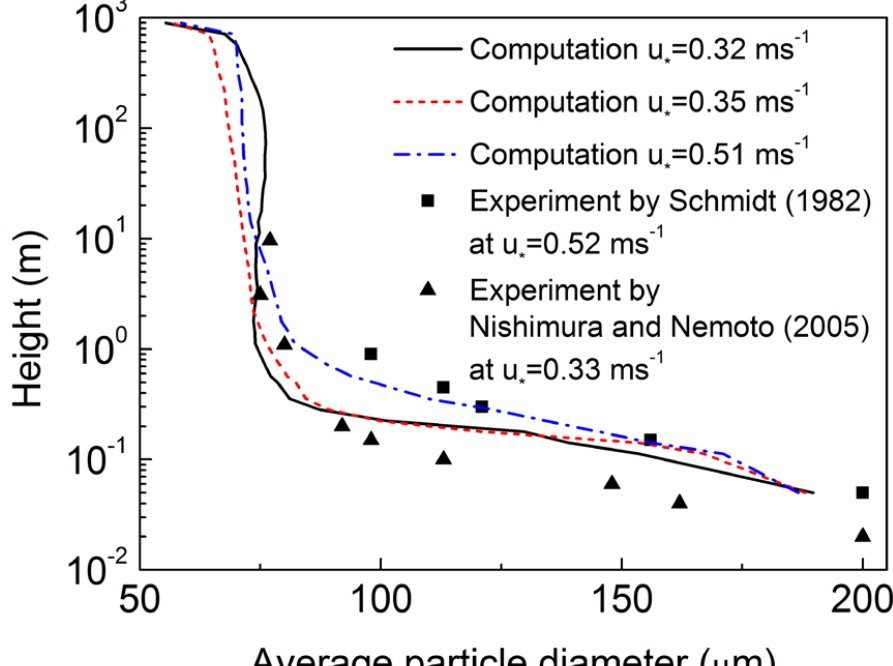


**Figure** 4. Variation of average particle diameter versus height.
Then, the particle size distributions at various heights are also compared with
experiment results. As shown in Fig. 5, the heights are 0.05 m, 0.5 m and 1 m. The



modeled particle size distributions at various heights are consistent with the
measurements (Nishimura and Nemoto, 2005; Schmidt, 1982). Therefore, the
established model is able to produce a large-scale drifting snow storm.

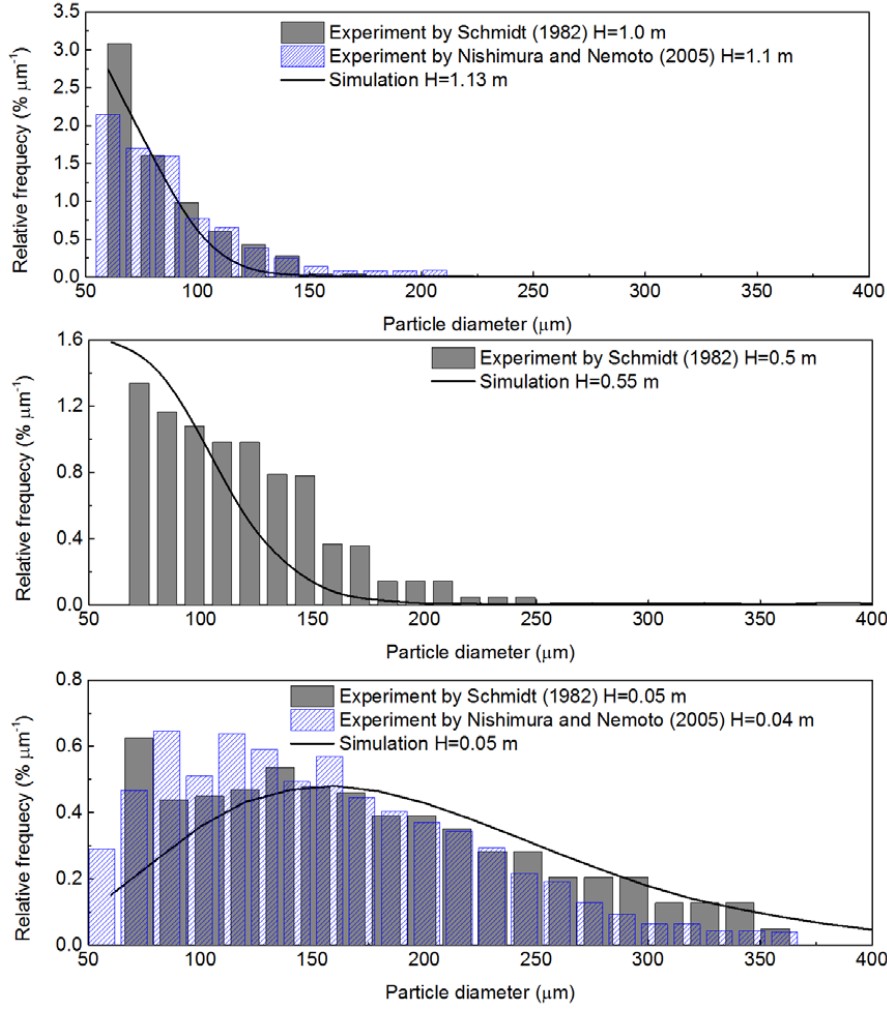


**Figure** 5. Particle size distribution at various heights.
**3.2  Snow transport flux**
The snow transport flux is of great importance to predict the mass and energy
balances of ice sheets. The total transport flux can be obtained from vertical



integration of the snow transport flux profile.

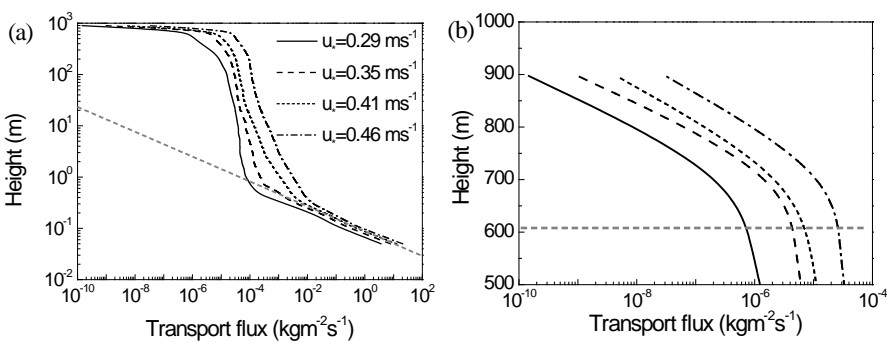


**Figure** 6. Variations of snow transport flux versus height.

The profiles of snow transport rate, per unit area, per unit time, under various

friction velocities are shown in Fig. 6(a). It can be seen that the transport flux
undergoes a sharp decrease with height at lower altitude (e.g., below 1.0 m), however,
the transport flux tends to decrease rather gentle until almost the top of the drifting
snow storm, as shown in Fig. 6(b), probably due to the large-scale turbulent motion
and increasing wind speed with height. In other words, the suspension flux of drifting
snow at higher altitudes, previously not observed, may be much larger than we
previously thought.

In previous studies, the transport flux profile is commonly described using an

exponential decay form based on the extrapolation from measurements near the
surface (Mann et al., 2000; Nishimura and Nemoto, 2005; Schmidt, 1982; 1984;
Tabler, 1990), which may result in a considerable underestimate of the total transport
flux. The proportions of suspension flux above a given height $h_c$ (referred as $Q_c$) to
the total suspension flux $Q_s$ are shown in Fig. 7, in which snow particles below 0.1
m are not calculated (Mann et al., 2000).

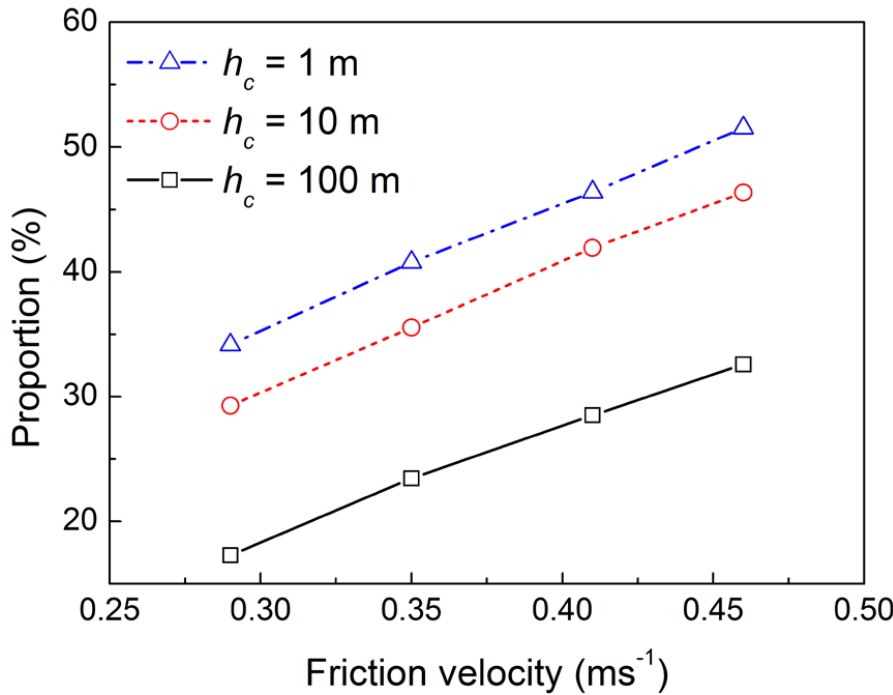


**Figure** 7. Proportion of suspension flux above $h_c$ to the total suspension flux under
various friction velocities.
From Fig. 7, the contribution of $Q_c$ to the total suspension flux is non-negligible
under various $h_c$, the proportion of $Q_c$ when $h_c$ =100 m to the total suspension flux
has exceeded 30% when the friction velocity is 0.46 ms$^{-1}$. At the same time, the
proportion of $Q_c$ to the total suspension flux increases with friction velocity but
decreases with increasing $h_c$.
In this way, not only the snow transport flux, but also the sublimation of
suspended snow particles should be reevaluated because the sublimation rate of snow
particles higher in the air may be much larger than near the surface due to the lower
air humidity and greater wind speed at higher altitude (Mann et al., 2000; Nishimura
and Nemoto, 2005; Schmidt, 1982; 1984; Tabler, 1990).





### 3.3  Structures in a drifting snow storm


In a drifting snow storm, particles aggregate locally and produce special spatial
structures (as shown in Fig. 2). These structures should be directly related to the
turbulence structures present in the atmospheric boundary layer. Drifting snow storms
without atmospheric turbulence are shown in Fig. 8. Compared with Fig. 2, drifting
snow particles mainly travel at the near surface with a uniform spatial distribution
when atmospheric turbulence is not included.

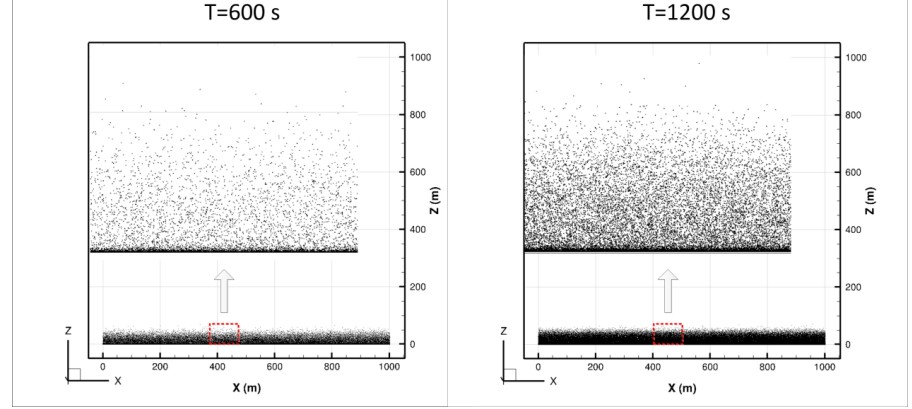


**Figure** 8. Drifting snow storm without atmospheric turbulence under friction velocity
of 0.35 ms$^{-1}$.

It is known that snow particles will become suspended if the local vertical wind

speed exceeds the terminal velocity of particle. In a turbulent atmospheric boundary
layer, there exists a large amount of turbulent structures with different scales and
shapes. The vertical wind speed component of large-scale turbulence (namely, updraft)
plays an important role in carrying snow particles to high altitude, while small scale
turbulence (e.g., the SGS fluctuating velocity) tends to spread particles from high
concentration zones to low concentration zones. As shown in Fig. 9(a), at the initial




period of a drifting snow storm, the structures in the drifting snow storm are
consistent with large-scale updrafts, and snow particles are mainly located in the
updraft. With the further development of the drifting snow storm, as shown in Fig.
9(b), more snow particles are scattered around the updraft bubbles although high
concentration particle clouds are still in the wind bubbles. When drifting snow storm
approaches its saturated state, snow particle clouds are almost connected together with
numerous high concentration zones inside.

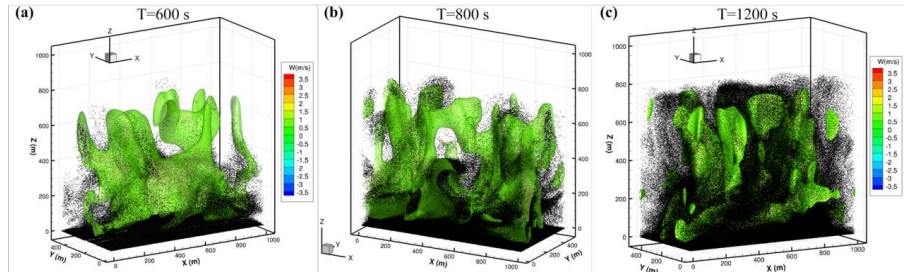


**Figure** 9. Evolution of drifting snow storm and vertical wind speed bubbles under
friction velocity of 0.35 ms$^{-1}$, and wind bubbles are iso-surface of vertical wind speed
with a value of 1.0 ms$^{-1}$.

The evolution of the depth of drifting snow storm can be divided into three typical

stages. In sequence, these phases are the rapid growth phase, the gentle growth stage,
and an equilibrium state, as shown in Fig. 10. Here, the depth of drifting snow storm
refers to the average height of the topmost particle during this period (100 s). The
rapid growth stage is mainly driven by large-scale turbulent motion, while the
turbulent diffusion by the SGS fluctuating velocity is the main contributor to the
gentle growth stage. The duration of second stage decreases with increasing friction
velocity, which mainly comes from the stronger turbulent diffusion under larger

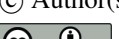



friction velocities.

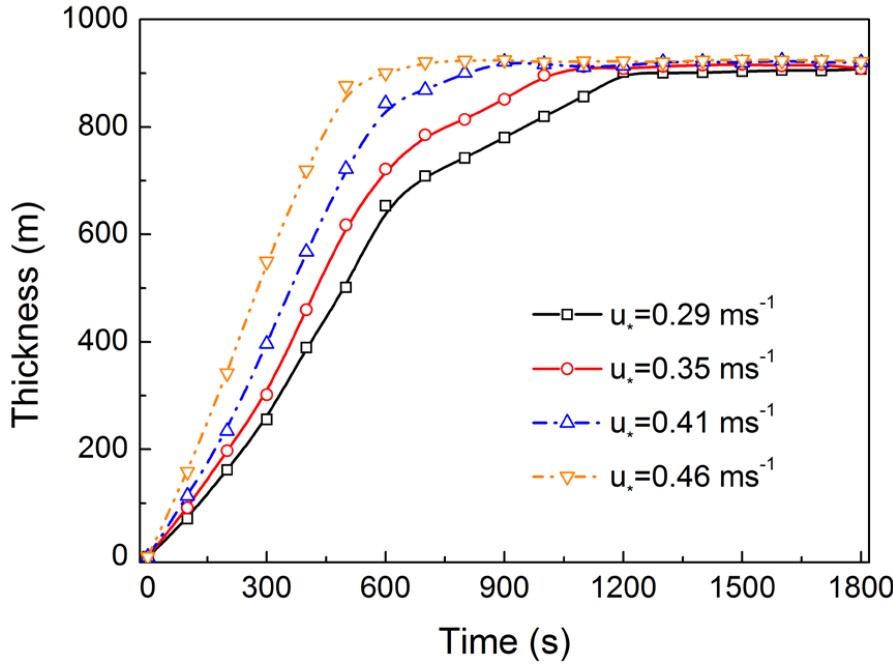


**Figure** 10. Time evolutions of the thickness of drifting snow storm under various
friction velocities.

At the same time, the time required for the drifting snow storm to reach its

maximum thickness decreases with friction velocity, ranging from about 1200 s to
approximate 600 s when the friction velocity increases from 0.29 $ms^{-1}$ to 0.46 $ms^{-1}$.
The thickness of saturated drifting snow storms is almost constant with a value
approximately 900 m under different friction velocities, probably because the
boundary layer depth as well as the surface heat flux are unchanged. Thus, the final
thickness of a drifting snow storm should be largely dependent on the maximum
height of atmospheric turbulences.
**4    Conclusion**



In this work, large-scale drifting snow storms are simulated in a large eddy simulation
combined with a particle tracking model that includes subgrid scale velocity
fluctuations. A typical drifting snow storm of several hundred meters in depth is
generated, and the structure of the particle cloud with different concentrations is also
produced. The transport flux profile has obviously different slopes near the surface
compared to higher altitudes, that is, transport flux at near surface decreases with
height sharply, but decreases more gentle at higher altitude. Previous studies may
largely underestimate the total transport during drifting snow storms.
At the same time, the evolution of the thickness of drifting snow storm generally
contains three stages. Drifting snow storm development generally begins with a rapid
growth stage driven by the large scale atmospheric turbulent motions, followed by a
gentle growth stage driven by the SGS fluctuating wind speed, before reaching an
equilibrium stage when the drifting snow approaches a saturated state. The second
stage becomes shorter with increasing friction velocity, mainly because stronger
turbulence under higher friction velocity enhances the turbulent diffusion of particles.

*Acknowledgements.* This work is supported by the CARDC Fundamental and Frontier
Technology Research Fund (FFTRF-2017-08, FFTRF-2017-09), the State Key
Program of National Natural Science Foundation of China (91325203), the National
Natural Science Foundation of China (11172118, 41371034), and the Innovative
Research Groups of the National Natural Science Foundation of China (11121202),
National Key Technologies R & D Program of China (2013BAC07B01).



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
