# Peer review of "A simulation of the large-scale drifting snow storm in a turbulent"

_The Cryosphere, 2018_

## Referee Comment (RC1) · Anonymous Referee #1 · 14 Aug 2018

In this manuscript, the authors used the large eddy simulation combined with the Lagrangian particles motion model to calculate the large-scale drifting snow storm. While their basic idea is interesting, the paper needs a revision before been published. The points of criticism are discussed in more detail in the following. (1) The author simulates the drifting snow storm in the manuscript. What are the differences between the drifting snow storm and the general blowing snow on the physical mechanism? How is it reflected in the model of this manuscript? (2) The mesh size set in this manuscript is much larger than the particle size. How do you determine the wind speeds of the particles position when calculating the particles motion? (3) The author mentions that a particle represents one particle parcel in Section 2.4. How many particles does the

particle parcel contain? What is the time step for calculating the particles? (4) The author mentions that the bottom boundary condition of the particles is calculated by Section 2.3, but Equation 12 shows that the impact and lift-off particles are the same, how does the particle in the air increase? (5) The author cites the work of Vinkovic et al. (2016) in Equation 4. The SGS velocity in the work of Vinkovic et al. (2016) is attached to the solid particles, but the author seems to attach it to the flow field. Why? (6) The result that the proportion of particles below 100 $\mu$m in the particle size distribution at 0.05 m in Figure 5 of this paper is obviously smaller than that of the experimental results. Why? (7) Figure 6a shows that the rate of snow transport flux has a mutation at 1 m, while the rate of the average particle size of snow particles in Figure 4 also has a mutation at 1 m. Is there any relationship between them? (8) Figure 10 shows that the thickness of drifting snow storm eventually developed to about 900m. Is this because the author set the upper boundary to 1000m? If the upper boundary is set higher, will the thickness of drifting snow storm continue to increase? (9) The author mentions that the particles enter the high-altitude causing by large-scale turbulence structure. Therefore, the authors show the distribution of airborne particles with and without consideration of atmospheric turbulence in Figure 2 and Figure 8 respectively. What are the differences between the two examples in Figure 2 and Figure 8 when calculating the flow field? What equations are used to calculate atmospheric turbulence? In addition, the author should give a comparison of the flow field structure in these two cases, so that the readers can understand this part of the content more clearly. (10) The author gives the vertical wind speed bubbles (1 m/s) in Figure 9, indicating that the particles are substituted into the upper air by the ascending airflow. Why use a 1m/s here? Is it the critical speed at which the particles become suspended particles? (11) There are some writing errors in this manuscript. For example, 'is' should be changed to 'are' in line 313 of page 19.

---

## Referee Comment (RC2) · Anonymous Referee #2 · 1 Oct 2018

General Comments :

The submitted manuscript described novel large-eddy simulations of large-scale blowing snow-storms. While the models utilized are well-established, such a phenomenon has not been previously explored using LES. The results of the simulations and their description and analysis are interesting and this reviewer feels that this study may be published in TC.

However, there are some major concerns that should be addressed before hand. The comments are listed below ordered by section.

Specific comments :

[Figure]

Section 2.1 : There seems to be misunderstanding about the use of the SGS velocity approach of Vinkovic et al. The SGS velocity is defined with respect to the frame of reference of the particle and not the flow. Thus, the splitting of local wind velocity as 'large-scale' and 'subgrid-scale' computed using Eq. 4 is incorrect.

Section 2.3 : Note that $\tau$ is not the total fluid shear stress but the total shear stress. When there are negligible particles, say at z > 1 m, $\tau$ and $\tau_f$ are equal. In lines 148-149, why is the ejection number set to 1 ? where does this value come from ? Sugiura and Maeno measured a much higher value .

Section 2.4 : Why is the initial potential temperature and relative humidity of the atmosphere described ? Is it relevant for the discussion ?

Section 2.4: The imposition of constant heat flux at the surface is perhaps the most questionable point for this reviewer. The study of Pomeroy and Essery found the 50 W/m2 flux for a brief period of time ( 20 mins perhaps ) during which, there was no blowing snow. Infact for most of the study, the sensible heat flux is either negligible or negative. The imposition of a constant heat flux at the surface is in effect creating a convective boundary layer that is providing a constant supply of energy in the form of vertical motions.

Section 2.4: line 179: How many snow particles are present in one particle parcel ?

Section 2.4: What is simulation time step for the flow as well as for the particle dynamics ?

Section 3.1 : This reviewer ( as well as the readers !) would highly appreciate vertical profiles of horizontal wind speeds simulated for different $u_\star$.

Section 3.2 : lines 250- 253 : The exponentially decaying transport flux profile is used to describe the saltation layer only and not the suspension layer.

Figure 7 and the corresponding text is a good result - but how are these numbers affected by the surface heat flux imposed ?

Section 3.3 : Lines 273-274 and Figure 8 : what is meant by snow storms without atmospheric turbulence ? How was this simulation achieved ? This is extremely unclear.

Section 3.3 : Figure 10 and the corresponding text : This reviewer feels that this result is extremely dependent on the imposed heat flux at the surface – How is this 'thickness' dependent of the surface heat flux ? The snow particles in the present case seem to reach the top of the computational domain !

---

## Author Comment (AC1) · 15 Oct 2018

Thanks for your relevent comments and suggestions. According to your comments, we have made a substantial revision to the original manuscript such that a clear description on the research is displayed in the revised manuscript (the directly changes can be seen in the Revised manuscript in tracking form). The detailed responses to comments of referees are in the Response to reviewer1.

Please also note the supplement to this comment:
https://www.the-cryosphere-discuss.net/tc-2018-134/tc-2018-134-AC1-supplement.zip

---

## Author Comment (AC2) · 22 Oct 2018

Thank you very much for your relevent comments and suggestions. According to your comments, we have made a substantial revision to the original manuscript such that a clear description on the research is displayed in the revised manuscript (the directly changes can be seen in the Revised manuscript in tracking form). The detailed responses to all the comments are in the Response to reviewer 2.

Please also note the supplement to this comment:
https://www.the-cryosphere-discuss.net/tc-2018-134/tc-2018-134-AC2-supplement.zip

---

## Author Response (AR1)

**Authors' Responses to the Comments on the Manuscript "A simulation of the large-scale drifting snow storm in a turbulent boundary layer"**

**General Response to the Comments:**

According to your comments, we have made a substantial revision to the original manuscript such that a clear description on the research is displayed in the revised manuscript (the directly changes can be seen in the revised manuscript with changes highlights). The detailed responses to comments of referees are as follows (see blue part in this reply):

**Responses to Comments of Reviewer#1:**

**General comments:**

**[Comment]** In this manuscript, the authors used the large eddy simulation combined with the Lagrangian particles motion model to calculate the large-scale drifting snow storm. While their basic idea is interesting, the paper needs a revision before been published. The points of criticism are discussed in more detail in the following.

**[Response]** Thanks for your careful reviews. A substantial revision to the original manuscript has been made according to your kind advice as listed in specific comments, as shown in the following responses.

**Specific comments:**

**[Comment 1]** The author simulates the drifting snow storm in the manuscript. What are the differences between the drifting snow storm and the general blowing snow on the physical mechanism? How is it reflected in the model of this manuscript?

**[Response 1]** Thanks for your this recommendations. The general blowing snow model pays attention to the particle motions at the near surface, and typically includes four sub-processes: aerodynamic entrainment, grain-bed collision, particle trajectory

and wind modification (Nemoto and Nishimura, 2004). However, the key physical process for a drifting snow storm is the particle's motion in atmospheric turbulences (especially the large-scale coherent structures), and a reasonable bottom boundary condition for particles is the basic.

From the view point of model, one the one hand, the three-dimensional large eddy simulation model combined with a proper model setting is necessary to produce large scale turbulent structures; on the other hand, a steady-state saltation condition is needed for the development of the drifting snow storm.

In the revised manuscript, the description 'The large-scale drifting snow storm is produced and its spatial structures and transport features are analyzed.' has been modified into 'The large-scale drifting snow storm is produced under the actions of large-scale turbulent structures combined with a steady-state snow saltation boundary condition for particles, and its spatial structures and transport features are analyzed.', as shown in line 69-72.

**[Comment 2]** The mesh size set in this manuscript is much larger than the particle size. How do you determine the wind speeds of the particles position when calculating the particles motion?

**[Response 2]** Thanks for your comment. In the process of calculating the particle's motion, the wind speed component at the particle's position is determined by the wind speeds at surrounding grid points through a linear interpolation algorithm. The sentence 'in which  $\tilde{u}_i(\vec{x}(t))$  is determined by the wind speeds of surrounding grid points through the linear interpolation algorithm' has been added in line 130-132 of the revised manuscript.

[Comment 3] The author mentions that a particle represents one particle parcel in Section 2.4. How many particles does the particle parcel contain? What is the time step for calculating the particles?

**[Response 3]** Thanks for your comment. We use one particle parcel to represents 2.5e7 snow particles. The description 'In this simulation, each particle parcel contains  $10^7$  snow particles.' has been added in line 214-215 of the revised manuscript.

At the same time, the particle time step is determined by the minimum of particle relaxation time  $T_p = \rho_p d_p^2 / 18\rho v$  to ensure a smooth particle trajectory (Dupont et al., 2013). The description 'The large time step and small time step (acoustic wave integral) for the wind field calculation are 0.1 s and 0.02 s, respectively, and the particle time step is determined by the minimum of particle relaxation time.' has been added in line 215-217 of the revised manuscript.

[**Comment 4**] The author mentions that the bottom boundary condition of the particles is calculated by Section 2.3, but Equation 12 shows that the impact and lift-off particles are the same, how does the particle in the air increase?

**[Response 4]** Thanks for your careful reviewing. The steady-state saltation is set as the bottom condition for snow particles. For a steady-state saltation, the impact and lift-off particles should be equivalent, thus, Equation (12) are used to guarantee a steady-state saltation throughout the calculation. In this condition, if some of the snow particles within the saltation layer are transported to higher in the air (the saltation layer becomes undersaturated), more particles will lift-off from the surface to replenish the saltation layer until a saturated state is reached.

In order to make it more clearly, the descriptions 'In this condition, if some of the snow particles within the saltation layer are transported to higher in the air by turbulent vortexes (the saltation layer becomes undersaturated), more particles will lift-off from the surface to replenish the saltation layer until a saturated state is reached.' are added in line 184-187 of the revised manuscript.

[**Comment 5**] The author cites the work of Vinkovic et al. (2016) in Equation 4. The SGS velocity in the work of Vinkovic et al. (2016) is attached to the solid particles, but the author seems to attach it to the flow field. Why?

**[Response 5]** Thanks for your comment. The subgrid scale (SGS) velocity is related to the local turbulent kinetic energy, but it has no any impacts on the wind field. Thus, the SGS velocity is attached to the solid particles essentially. In order to make it more clearly, the contents about SGS velocity have been moved to section 2.2, and the description 'Namely, the local wind velocity  $\tilde{u}_i(\vec{x}(t))$  is composed of a resolved

Eulerian large-scale part  $\tilde{u}_i(\vec{x}(t))$  (obtained from the linear weighting of surrounding grid points) and a fluctuating SGS contribution  $u'_i(t)$ ' has been changed into 'Namely, the local relative is expressed as  $V_{ri} = \tilde{u}_i(x_p) - u_{pi} + u'_i$ , in which  $\tilde{u}_i(\vec{x}_p)$  is the resolved large-scale wind speed at the particle's position and is determined by the resolved wind speeds of surrounding grid points through the linear interpolation algorithm.', as shown in line 129-132 of the revised manuscript.

[**Comment 6**] The result that the proportion of particles below 100 m in the particle size distribution at 0.05 m in Figure 5 of this paper is obviously smaller than that of the experimental results. Why?

**[Response 6]** Thanks for your careful reviewing. In Fig. 5 of the original manuscript, the proportion of particles below 100 m in the particle size distribution at 0.05 m is smaller than that of the experimental results. The reason could be that mid-air collisions, occurred frequently within the high concentration saltating snow cloud at the near surface, play an important role in conveying larger particles to high altitude (Carneiro et al., 2013). However, the effect of mid-air collision mechanism is beyond the scope of the current study.

In the revised manuscript, the description 'Besides, it can be seen that the proportion of particles below 100  $\mu$ m in diameter at 0.05 m is smaller than that of the experimental result. The reason could be that mid-air collisions, occurred frequently within the high concentration saltating snow cloud at the near surface, play an important role in conveying larger particles to higher altitude (Carneiro et al., 2013). However, the mid-air collision mechanism is beyond the scope of the current study.' has been added in line 273-278.

[**Comment 7**] Figure 6a shows that the rate of snow transport flux has a mutation at 1 m, while the rate of the average particle size of snow particles in Figure 4 also has a mutation at 1 m. Is there any relationship between them?

[**Response 7**] Thanks for your comment. Indeed, the snow transport flux profile is related to the average particle size profile. The transition of snow transport flux

profile at about 1 m should be caused by the different motion states of particles with different particle sizes. As shown in Fig. 4, the mean particle diameter decreases rapidly with height below the critical height of approximately 1 m, and almost keeps constant above this height. Above the critical height, the particle gravities and relaxation times are small, thus, particles follow the turbulent flow in the state of suspension. However, below this height, plenty of larger particles have much larger relaxation times and gravities, thus, there exist relative speed between particle and wind field because particle inertia plays an important role.

In the revised manuscript, the description 'Besides, the transition of snow transport flux profile at about 1 m should be mainly caused by the different motion states of particles with different particle sizes, as shown in Fig. 4. Above the critical height, particles generally follow the turbulent flow in the state of suspension because their gravities and relaxation times are small enough. However, plenty of larger particles at the near surface make the particles velocity differs from the wind speed, since particle inertia plays an important role.' has been added in line 298-303.

[**Comment 8**] Figure 10 shows that the thickness of drifting snow storm eventually developed to about 900m. Is this because the author set the upper boundary to 1000m? If the upper boundary is set higher, will the thickness of drifting snow storm continue to increase?

**[Response 8]** Thanks for your comment. Actually, the height of the domain is determined by a series of testing simulations with various domain heights. As shown in Fig. R1, under current model settings, the thickness of the fully developed turbulent boundary layer basically do not vary with the height of the domain. The reason could be that the turbulent boundary layer is a shear force dominated flow with constant initial boundary layer depth and the surface heat flux (Moeng and Sullivan, 1994). Drifting storm with different thicknesses may be achieved through changing the initial field and surface heat flux.

The description 'Higher domain heights are also tested with the same model settings, and the thickness of the drifting snow seems basically unchanged. Drifting

snow storm with difference thicknesses may be achieved by changing the initial state of the air and surface heat flux.' has been added in line 388-391 of the revised manuscript.

**Figure** R1. Iso-surfaces of vertical wind speed bubbles with a value of 1.0 ms-1 with different domain height (a)1.0 km and (b) 1.5 km. All simulation settings are the same for both simulations except the height of the domain.

[Comment 9-1] The author mentions that the particles enter the high-altitude causing by large-scale turbulence structure. Therefore, the authors show the distribution of airborne particles with and without consideration of atmospheric turbulence in Figure 2 and Figure 8 respectively. What are the differences between the two examples in Figure 2 and Figure 8 when calculating the flow field? What equations are used to calculate atmospheric turbulence?

**[Response 9-1]** Thanks for your comment. First of all, the atmospheric turbulence is calculated by the large eddy simulation model (Equation 1~3) through wind shear combined with a small heat flux at the bottom (Moeng and Sullivan, 1994). Then, the only difference between the two examples in Fig. 2 and Fig. 8 is that the resolved wind speed at particle's position  $(\tilde{u}_i(\vec{x}_p))$  in Fig. 8 is replaced by a given value obtained from the standard logarithmic profile during calculating particle's trajectory. In this way, the effect of resolved large-scale turbulent structures on the development of the drifting snow storm can be removed from the example in Fig. 8.

In the revised manuscript, the description 'This simulation is achieved by

replacing the resolved wind speed at particle's position  $(\tilde{u}_i(\vec{x}_p))$  with a given value obtained from the standard logarithmic profile, and the other model settings and simulation procedures stay the same with other simulations. In this way, the effect of large-scale turbulent structures on the development of the drifting snow storm vanishes.' has been added in line 339-344.

[Comment 9-2] In addition, the author should give a comparison of the flow field structure in these two cases, so that the readers can understand this part of the content more clearly.

**[Response 9-2]** Thanks for your suggestion. As discussed in [Response 9-1], the flow field structures in these two cases are the same. However, in order to make the this part of the content more clearly, the description 'This simulation is achieved by replacing the resolved wind speed at particle's position  $(\tilde{u}_i(\vec{x}_p))$  with a given value obtained from the standard logarithmic profile, and the other model settings and simulation procedures stay the same with other simulations. In this way, the effect of large-scale turbulent structures on the development of the drifting snow storm vanishes.' has been added in line 339-344 of the revised manuscript.

[**Comment 10**] The author gives the vertical wind speed bubbles (1 m/s) in Figure 9, indicating that the particles are substituted into the upper air by the ascending airflow. Why use a 1m/s here? Is it the critical speed at which the particles become suspended particles?

**[Response 10]** Thanks. The reviewer is right that the wind speed of 1m/s is approximately the critical speed at which the particles of mean particle size become suspended particles, because the maximum diameter of suspended particles is found to be approximately the mean particle size of the lift-off particles. The description '(corresponding to the critical wind speed at which the particle of mean particle size becomes suspended particle, since the maximum diameter of suspended particles is found to be approximately equals to the mean particle size of the lift-off particles)' has been added in line 367-370 of the revised manuscript.

[Comment 11] There are some writing errors in this manuscript. For example, 'is'

should be changed to 'are' in line 313 of page 19.

**[Response 11]** Thanks for your careful reviewing. The sentence 'The thickness of saturated drifting snow storms is almost constant with a value approximately 900 m under different friction velocities' has been changed into 'The thicknesses of saturated drifting snow storms are almost constant with a value approximately 900 m under different friction velocities' in line 386-387 of the revised manuscript.

**Responses to Comments of Reviewer#2:**

**General Comments :**

**[Comment]** The submitted manuscript described novel large-eddy simulations of large-scale blowing snow-storms. While the models utilized are well-established, such a phenomenon has not been previously explored using LES. The results of the simulations and their description and analysis are interesting and this reviewer feels that this study may be published in TC. However, there are some major concerns that should be addressed before hand. The comments are listed below ordered by section.

**[Response]** Thanks for your careful reviews and relevant comments. A substantial revision to the original manuscript has been made according to your kind advice as listed in specific comments, please see our point-to-point response below.

**Specific comments:**

**[Comment 1]** Section 2.1 : There seems to be misunderstanding about the use of the SGS velocity approach of Vinkovic et al. The SGS velocity is defined with respect to the frame of reference of the particle and not the flow. Thus, the splitting of local wind velocity as 'large-scale' and 'subgrid-scale' computed using Eq. 4 is incorrect.

**[Response 1]** Thanks for your relevant comment. In order to correct this mistake, the sentences 'Namely, the local wind velocity  $\tilde{u}_i(\vec{x}(t))$  is composed of a resolved Eulerian large-scale part  $\tilde{u}_i(\vec{x}(t))$  (obtained from the linear weighting of surrounding grid points) and a fluctuating SGS contribution  $u'_i(t)$ .' have been

changed into 'Namely, the local relative is expressed as  $V_{ri} = \tilde{u}_i(x_p) - u_{pi} + u'_i$ , in which  $\tilde{u}_i(\vec{x}_p)$  is the resolved large-scale wind speed at the particle's position and is determined by the resolved wind speeds of surrounding grid points through the linear interpolation algorithm.' in line 129-132 of the revised manuscript. Besides, the contents about SGS velocity have been moved to section 2.2 of the revised manuscript for a better understanding.

**[Comment 2]** Section 2.3 : Note that  $\tau$  is not the total fluid shear stress but the total shear stress. When there are negligible particles, say at z > 1 m,  $\tau$  and  $\tau_f$  are equal. In lines 148-149, why is the ejection number set to 1 ? where does this value come from ? Sugiura and Maeno measured a much higher value.

**[Response 2]** Thanks for your careful reviewing. According to your comment, the expression 'total fluid shear stress' has been modified into 'total shear stress' throughout the revised manuscript.

On the other hand, the splash model of Sugiura and Maeno (2000) determines the relation between the ejection number and the speed and incident angle of the impactor, and the ejection number includes both rebound and ejected particles. They measured a much higher ejection number during the development of the drifting snow. However, we set a saturated saltation layer as the bottom boundary condition for particles, in which case the numbers of impact and lift-off particles should be equivalent (one impactor corresponds one ejected particle). Thus, the ejection number of 1 comes from the steady saltation condition.

In order to make it more clearly, the description 'and  $\langle v_i \rangle$  is set to be the threshold of impact velocity, which is determined by setting ejection number  $n_e = 0.51 v_i^{0.6} \theta_i^{0.16}$  equal to 1.' has been modified into 'and  $\langle v_i \rangle$  is set to be the threshold of impact velocity. Considering the steady-state saltation condition (one impact particle generates one ejecta on average),  $\langle v_i \rangle$  is determined by setting ejection number  $n_e = 0.51 v_i^{0.6} \theta_i^{0.16}$  equal to 1.' in the revised manuscript, as shown in

**line 170-172.**

[Comment 3] Section 2.4 : Why is the initial potential temperature and relative humidity of the atmosphere described ? Is it relevant for the discussion ?

**[Response 3]** Thanks for your careful reviewing. As a matter of fact, the initial potential temperature and relative humidity of the atmosphere are used to determine the air density. In the revised manuscript, the content ' $\rho = p(1-q_v/(\varepsilon+q_v))(1+q_v)/(R_dT)$  is the air density, in which p,  $q_v$ , R and T are the pressure, the specific humidity, the gas constant (287.0  $Jkg^{-1}K^{-1}$ ) and temperature of the air, respectively, and  $\varepsilon$ =0.622 is a constant.' has been added in line 84-86 of the revised manuscript. **[Comment 4]** Section 2.4: The imposition of constant heat flux at the surface is perhaps the most questionable point for this reviewer. The study of Pomeroy and Essery found the 50 W/m2 flux for a brief period of time ( 20 mins perhaps ) during which, there was no blowing snow. In fact for most of the study, the sensible heat flux

is either negligible or negative. The imposition of a constant heat flux at the surface is in effect creating a convective boundary layer that is providing a constant supply of energy in the form of vertical motions.

**[Response 4]** Thanks for this relevant comment. Typically, the atmospheric turbulence is generated and maintained by two forces: wind shear and buoyancy force. Most studies set the heat flux to zero, which corresponds to an ideal shear-driven planetary boundary layer (PBL). However, these two forces may act together to modify the flow field in actual situations (Moeng and Sullivan, 1994). In this study, a small heat flux is added in the shear-dominated PBL to produce a 'intermediate PBL' that is closer to the real situation (A buoyancy-dominated convective PBL generally requires a heat flux larger than 200 W/m2). Although the surface heat flux may be changed during drifting snow, however, the smaller surface heat flux basically not affect the structures of drifting snow storms, also see the analysis in [Response 9] and [Response 11].

In order to make it more clearly, the description 'Actually, this condition

corresponds to a 'intermediate' turbulent boundary layer that dominated by wind shear force. Thus, the structures of the drifting snow storm should not be affected by the changing surface heat flux significantly if the surface heat flux is small. Further simulations with different values of surface heat flux (

Figure R2. Horizontal wind speed profiles under various friction velocities.

In the revised manuscript, Fig. R2 and the description 'The mean horizontal wind speed profiles of the fully developed turbulent boundary layer under various friction velocities are shown in Fig. 7b. The horizontal wind speed increases with height and changes into a constant above the boundary layer. The rapid decrease of the snow transport flux occurs at about the top of the turbulent boundary layer, mainly because turbulences become weaker above this height and less particles can be transported to a higher altitude.' have been added, as shown in line 292-297 and Fig. 7.

**[Comment 8]** Section 3.2 : lines 250- 253 : The exponentially decaying transport flux profile is used to describe the saltation layer only and not the suspension layer.

[**Response 8**] Thanks for your careful reviewing. According to your comment, the sentence 'In previous studies, the transport flux profile is commonly described using an exponential decay form based on the extrapolation from measurements near the surface (Mann et al., 2000;Nishimura and Nemoto, 2005;Schmidt, 1982, 1984;Tabler, 1990), which may result in a considerable underestimate of the total transport flux.' has been modified into 'In previous studies, only the transport fluxes at the near surface are commonly measured (Mann et al., 2000;Nishimura and Nemoto, 2005;Schmidt, 1982, 1984;Tabler, 1990), thus, the features of the entire transport flux profile is largely unclear, which may result in considerable uncertainties about the total transport flux.' in the revised manuscript, as shown in line 307-312.

[Comment 9] Figure 7 and the corresponding text is a good result - but how are these

numbers affected by the surface heat flux imposed ?

**[Response 9]** Thanks. According to your comment, the effect of surface heat flux  $q_s$  on the structures of drifting snow storm is examined. The results indicate that the structures of drifting snow storms are less affect by the surface heat flux when it is small (e.g.,  $q_s \leq 100 \text{ Wm}^{-2}$ ). As shown in Fig. R3, the proportion of the suspension flux above  $h_c$  to the total suspension flux is only slightly affected by the surface heat flux, and the influence of surface heat flux becomes weaker and weaker with the increasing friction velocity, mainly because larger friction velocity results in stronger turbulence under the actions of wind shear.

---

## Author Response (AR2)

**Authors' Responses to the Comments on the Manuscript**

**"A simulation of the large-scale drifting snow storm in a turbulent boundary layer"**

**General Response to the Comments:**

According to your comments, we have modified the original manuscript carefully (see the point-by-point reply to the comments, and the marked-up manuscript below). We have also checked the manuscript carefully for typos, authors and corresponding affiliations, terminology, variables in equations, acknowledgements and references.

**Responses to Comments of Editor:**

**[Comment]** Your revised version of manuscript will be accepted for publication in TC after some minor revisions are made. Please see the comments from the reviewer below. At this point I ask you to carefully proofread the manuscript and make any necessary correction, such as missing units etc.

**[Response1]** Thank you for your recommendation.

We have carefully proofread the manuscript according to your and reviewer comments. The missing units have been added, and the typos, acknowledgements, variables in equations and references have been updated in the revised manuscript.

**Responses to Comments of reviewer:**

**[Comment]** The authors have improved the manuscript and mostly taken into account the previous remarks. I find that this work gives a good basis for further research in the area. Therefore, the manuscript now deserves publication. However, there are few mistakes in the manuscript, such as there is not unit after ' '.

**[Response1]** Thanks for your careful reviewing.

According to your comment, the missing units have been added in the revised manuscript. At the same time, we have also checked the manuscript carefully, the acknowledgements, typos, variables in equations and references have been updated in the revised manuscript.

Finally, once again we appreciate you for your good and comprehensive comments. Those revisions according to your comments really make this manuscript improve a lot.
Thank you!

Yours sincerely,
Zhengshi Wang, and Shuming Jia.

**A list of changes**

1. The word 'sources' has been modified into 'source' in line 27.

2. The symbol '$R$' has been modified into '$R_d$' in line 84.

3. The expression '$v = 1.5e - 5$' has been modified into '$v = 1.5 \times 10^{-5} \quad m^2 s^{-1}$' in line 109.

4. The word 'dynamic' has been modified into 'kinematic' in line 109.

5. The symbol '$d$' has been modified into '$\bar{d}_p$' in equation (10).

6. The sentence '$g$ is the gravity acceleration' in line 139 has been deleted.

7. The expression '$1000 \times 500 \times 1000$ m' has been modified into '1000 m$\times$500 m$\times$1000 m' in line 175.

8. The expression '$d$ is the particle diameter, and' in line 210 has been deleted.

9. The symbols '$d$, $\alpha$ and $\beta$' has been modified into '$d_p$, $\alpha_p$ and $\beta_p$', respectively, in equation (17).

10. The symbols '$\alpha$' and '$\beta$' has been modified into '$\alpha_p$' and '$\beta_p$' in line 208 and 210.

[revised manuscript text omitted]